# A Novel Dipeptidyl Peptidase-4 Inhibitor DA-1229 Ameliorates Tubulointerstitial Fibrosis in Cyclosporine Nephrotoxicity in Mice

**DOI:** 10.3390/life11030251

**Published:** 2021-03-18

**Authors:** Hye Sook Min, Ji Eun Lee, Jung Yeon Ghee, Young Sun Kang, Jin Joo Cha, Jee Young Han, Sang Youb Han, Dae Ryong Cha

**Affiliations:** 1Department of Internal Medicine, Division of Nephrology, Wonkwang University, 460 Jeollabuk-do 54538, Korea; glorymhs@naver.com (H.S.M.); borisoo12@hanmail.net (J.E.L.); 2Department of Internal Medicine, Korea University Ansan Hospital, 516 Kojan-Dong, Ansan City 425-020, Korea; gheejy@nate.com (J.Y.G.); starch70@korea.ac.kr (Y.S.K.); minipearl@korea.ac.kr (J.J.C.); 3Department of Pathology, Inha University, 197 Inje-ro, Gimhae-si 50834, Korea; jeeyhan@inha.ac.kr; 4Department of Internal Medicine, Division of Nephrology, Inje University, 170 Goyang 10380, Korea; hansy@paik.ac.kr

**Keywords:** cyclosporine nephrotoxicity, DPP-4, DA-1229, inflammation, fibrosis, oxidative stress

## Abstract

Cyclosporine A (CyA) is an immunosuppressive agent that induces nephrotoxicity with long-term treatment. The roles of DPP-4 and its inhibitors in cyclosporine nephrotoxicity are not fully understood. Therefore, we investigated the effects of a novel DPP-4 inhibitor, DA-1229, on the progression of renal disease in an experimental cyclosporine nephrotoxicity model. Chronic cyclosporine nephrotoxicity was induced in six-week-old male ICR mice by subcutaneous injections of CyA at a dose of 30 mg/kg for four weeks. Animals were treated with DA-1229 at a dose of 300 mg/kg per day in food for four weeks. Although DPP-4 activity did not increase in the kidneys of mice with induced cyclosporine nephrotoxicity, DA-1229 treatment significantly suppressed DPP-4 activity in both plasma and renal tissues. DPP-4 inhibition by DA-1229 led to significantly decreased albuminuria and urinary excretion of 8-isoprosatane. DPP-4 inhibition also substantially suppressed pro-inflammatory effects, profibrotic molecules, and macrophage infiltration, and led to the improvement in renal structural changes. Our results suggest that DPP-4 inhibition by DA-1229 provides renoprotective effects in an animal model of cyclosporine nephrotoxicity via antioxidant, anti-inflammatory, and anti-fibrotic mechanisms. DPP-4 inhibition may be a useful new therapeutic approach for the management of progressive renal disease in cyclosporine nephrotoxicity.

## 1. Introduction

Dipeptidyl peptidase-4 (DPP-4) inhibitor is well known antidiabetic drug that enhances the levels of incretin hormones such as GLP-1 and GIP. Currently, several DPP-4 inhibitors, such as sitagliptin, vildagliptin, saxagliptin, linagliptin, and alogliptin are widely used in the treatment of type 2 DM [1,2]. However, whether the effects of DPP-4 inhibitors on diabetes are mediated solely through GLP-1 remains debatable. DPP-4 inhibitors lead to only minor increases in endogenous GLP-1. The reason for the different effects of DPP-4 inhibitors and GLP-1 is that DPP-4 does not only inactivate GLP-1 but also cleaves other substrates.

DPP-4 acts on various types of substrates, such as regulatory peptides, neuropeptides, and chemokines that are related to metabolism, glucose regulation, inflammation, cell migration, and cell differentiation [1,3]. DPP-4 is broadly expressed in multiple tissues including heart, lung, brain, liver, pancreas, vessels, intestine, prostate, uterus, thymus, lymph nodes, spleen, adipose tissue, and kidney, the last of which expresses the highest levels of DPP-4. Moreover, DPP-4 acts as a biochemical messenger in a wide range of cells including epithelial, endothelial, and various types of immune cells [1,4]. DPP-4 is especially closely related to the immune system and controls immune regulation in autoimmune diseases by regulating T-cell activity, migration, and proliferation [5,6].

In the kidney, DPP-4 expression is localized to the brush border membrane of the proximal tubules, the endothelial cell of glomeruli, and podocyte cells in Bowman’s capsule [1]. Previous experimental studies have shown that DPP-4 inhibitors have renoprotective effects in various models of diabetic and non-diabetic kidney disease independent of its hypoglycemic effects. DPP-4 inhibitors are associated with reduced albuminuria, and suppression of the inflammatory markers and oxidative stress in a streptozotocin-induced diabetic animal model [7,8]. The DPP-4 inhibitor ameliorated renal ischemic-reperfusion injury due to protection against apoptotic, inflammatory, and oxidative injuries [9]. In a previous study using a UUO mouse model, we suggested that the DPP-4 inhibitor protects against renal injury by several mechanisms associated with fibrosis, inflammation, and oxidative damage [10]. There are many previous reports suggesting that DPP-4 inhibition improves oxidant stress and renoprotective effects [11,12,13].

Although chronic antibody-mediated rejection is the leading cause of kidney allograft failure, calcineurin inhibitor toxicity is one of the important causes of allograft failure. Cyclosporin A (CyA) is an immunosuppressive agent that is commonly used to prevent allograft rejection in various organ transplantation and to slow the progression of various autoimmune diseases. CyA inhibits the expression of interleukin-2 in T lymphocytes through calcineurin inhibition [14]. However, nephrotoxicity is a significant side effect of long-term use of CyA. CyA induces overexpression of NAD(P)H oxidase in endothelial cells and overproduction of IL-6 and TGF-β1. CyA-induced nephropathy is related to inappropriate activation of the renin-angiotensin system (RAS), renal arteriolar vasoconstriction, arteriolar hyalinosis, tubulointerstitial fibrosis, tubular atrophy, and progressive renal injury [15,16]. Because DPP-4 is highly expressed and its activity is most abundant in the kidney, and DPP-4 is involved in inflammation and oxidative stress, DPP-4 may have an important pathogenic role in CyA nephrotoxicity. However, the actions of DPP-4 and its inhibitors in CyA nephrotoxicity are not fully understood. DA-1229 is a novel, potent, selective DPP-4 inhibitor [17]. Therefore, we investigated the role of a novel DPP-4 inhibitor, DA-1229, on the progression of renal disease in an experimental CyA-induced nephropathy model.

## 2. Materials and Methods

### 2.1. Animal Studies

Eight-week-old male ICR mice were divided into four experimental groups (*n* = 10 in each group): control group with vehicle injection, control group with DA-1229 treatment group, cyclosporine toxicity group, and cyclosporine toxicity group treated with DA-1229. All mice were fed a low salt diet (0.1% Na, Harlan Teklad Research Animal Feed). DA-1229 was administered at a dose of 300 mg/kg/day in chow admixture for four weeks and CyA was treated by subcutaneous injection with CsA (Chongkundang Pharm, Seoul, Korea) at 30 mg/kg/day for four weeks based on a previous study in mice [17,18]. DA-1229 tartrate, a DPPIV inhibitor, was obtained from Dong-A ST Research Institute (Yongin, Korea). All mice were provided with standard chow and food intake was checked to confirm the administered dose of DA-1229. Urinary albumin concentrations were measured using a competitive ELISA kit (ALPCO, Westlake, OH, USA). Urinary protein concentrations were determined using 30% TCA with the Bradford method and corrected by urine creatinine concentrations. Urinary 8-isoprostane levels were determined using an ELISA kit (Cayman Chemical, Ann Arbor, MI, USA). Serum creatinine levels were measured using a serum creatinine detection kit (Arbor Assays, MI, USA). Plasma active GLP-1 levels were determined by ELISA kit (Millipore, MO, USA). Systolic blood pressure was measured using tail-cuff plethysmography (Letica SA, Barcelona, Spain). Mice were euthanized under anesthesia through i.p. injections of sodium pentobarbital (50 mg/kg). All experiments were conducted in accordance with NIH guidelines and with the approval of the Korea University Institutional Animal Care and Use Committee (KOREA-2013-023).

### 2.2. DPP-4 Activity Assay

DPP-4 activity was determined in plasma and tissues as previously described [10]. Briefly, 0–40 μM of AMC (7-amino-4-methylcoumarin) standard was loaded into each well and read fluorometrically at Em/Ex = 360/465 nm, generating a linear standard curve. Next, 50 μL of plasma sample was added to each of the 96 wells, followed by the addition of 50 μL substrate solution (final concentrations 0.1 M HEPES, 50 μM Gly-Pro-AMC, and 50 μg/mL BSA) per well. Free AMC, which was released by DPP-4 activity, was quantitated by fluorometric measurements every 25 s for 300 s using a Spectramax GEMINI XPS microplate reader (Molecular Devices, Sunnyvale, CA, USA; Em/Ex = 360/465 nm, target temperature = 25 °C). To analyze DPP-4 activity in renal tissue, frozen tissue (10 mg) was homogenized in cold assay buffer (25 mM Tris-HCl, 140 mM NaCl, 10 mM KCl, pH 7.5, 0.1% BSA) and spun by centrifugation at 20,000× *g* for 20 min at 4 °C. DPP-4 activity in the lysates was measured as described above. DPP-4 activity in each tissue sample was expressed as the amount of cleaved AMC per minute per gram of tissue (μM/min/g tissue).

### 2.3. Histological and Immunohistochemical Analysis

Renal tissues were fixed in 4% paraformaldehyde and embedded in paraffin. Samples were cut into 4-μm slices and stained with periodic acid-Schiff and Masson’s trichrome. For immunohistochemical staining for α-SMA and F4/80, samples were microwaved for 10–20 min. To block endogenous peroxidase activity, 3.0% H_2_O_2_ in methanol was applied to the tissue sections for 20 min, then incubated for 60 min in 3% BSA/3% normal goat serum at room temperature. Then samples were incubated overnight at 4 °C with mouse monoclonal anti-F4/80 antibodies (1:100; Serotec Inc, Raleigh, NC, USA), rabbit polyclonal anti-α-SMA antibodies (1:100; Santa Cruz Biotechnology, Inc.). After overnight incubation, samples were incubated with secondary antibodies for 30 min. Immunoreaction was performed by incubation with a mixture of 0.05% 3,3-diaminobenzidine containing 0.01% H_2_O_2_ followed by counterstaining with Mayer’s hematoxylin. The degree of macrophage infiltration was counted and shown as the number of macrophages per high-power field. After α-SMA staining, the degree of tubulointerstitial fibrosis was assessed by a point-counting method as described previously [10].

### 2.4. Analysis of Gene Expression by Real-Time Quantitative PCR

Total RNA was extracted from renal cortical tissue using Trizol reagent and further purified using an RNeasy Mini kit (Qiagen, Valencia, CA, USA). The nucleotide sequences of all primers are shown in Appendix A. Real-time quantitative PCR was performed using a LightCycler 1.5 system (Roche Diagnostics Corp, Indianapolis, IN, USA) using SYBR Green technology. Thermocycling conditions consisted of 22–30 cycles of denaturation for 10 s at 95 °C followed by annealing and extension for 30 s at 60 °C.

### 2.5. Protein Extraction and Western Blot Analysis

Protein extracts were done from renal cortical tissues by a standard method, and a total of 40 μg of protein was electrophoresed on 10% SDS-PAGE minigels, and the membranes were incubated in blocking buffer overnight at 4 °C with rabbit polyclonal anti-TGFβ1 antibodies (1:200; Santa Cruz Biotechnology, Dallas, TX, USA), goat polyclonal anti-type I collagen α1 antibodies (1:2000, Abcam Plc, Boston, MA, USA), goat polyclonal anti-toll-like receptor 4 (TLR4) antibody (1:500; Santa Cruz Biotechnology), rabbit polyclonal anti-GLP1 receptor antibody (1:100, Abcam Plc), and mouse monoclonal anti-β-actin antibodies (1:5000, Sigma-Aldrich Corp, Saint Louis, MO, USA). The membranes were subsequently incubated with horseradish peroxidase-conjugated secondary antibodies (1:1000 dilution) for 60 min at room temperature. Immunoreactive bands were detected using enhanced chemiluminescence reagents (Amersham, Buckinghamshire, UK).

### 2.6. Statistical Analysis

Nonparametric analysis was performed due to the relatively small number of samples. The Kruskal–Wallis test was used to compare more than two groups, followed by the Mann–Whitney U-test, using SPSS for Windows 20.0 (SPSS, Chicago, IL, USA). *p*-values of 0.05 were considered statistically significant.

## 3. Results

### 3.1. Physical and Biochemical Parameters of Experimental Animals

Table 1 summarizes the physical and biochemical parameters obtained for each experimental group. There were no significant differences in body weight, food intake, water intake, urine volume, plasma levels of active GLP-1, or systolic blood pressure among groups. Serum creatinine was significantly increased in the CyA group compared to the vehicle group and markedly decreased in the DA-1229 treatment group compared to the CyA group.

### 3.2. Effects of DA-1229 on DPP-4 Activity in the Plasma and Kidney Tissue

DPP-4 activity in the plasma was not increased by CyA treatment, and DA-1229 treatment significantly suppressed plasma DPP-4 activity in both vehicle and CyA injection group 2.6–4.1s (Figure 1A). Renal DPP-4 activity did not show significant differences in mice that received CyA injections compared to the vehicle group, but mice treated with DA-1229 showed markedly suppressed DPP-4 activity in both vehicle and CyA injection groups (Figure 1B).

### 3.3. Effects of DA-1229 on Urinary Excretion of Protein, Albumin, and 8-Isoprostane

Urinary protein and albumin excretion were markedly increased after four weeks in the CyA injection group. DA-1229 treatment significantly decreased both protein and albumin excretion after four weeks in the CyA treatment group (Figure 2A,B). Urinary levels of 8-isoprostane were also markedly increased after four weeks in the CyA injection group, and treatment with DA-1229 abrogated urinary 8-isoprostane excretion after CyA injection (Figure 2C).

### 3.4. Effects of DA-1229 on Gene Expression Related to Inflammation and Fibrosis

We examined gene expression in kidney tissues to observe the anti-inflammatory and anti-fibrotic effects of DA-1229 treatment. Although there were no significant changes in PAI-1 and type IV collagen gene expression, expressions of CTGF, TGFβ1, and Type I collagen were significantly decreased by DA-1229 treatment in the CyA group (Figure 3A). In addition, the expressions of proinflammatory cytokines such as IL-1β, MCP-1, and TLR4 were significantly decreased in kidney tissue by DA-1229 treatment in the CyA group (Figure 3B). Western blot showed that expressions of Type I collagen, TGFβ1 and TLR4, and major fibrotic and inflammatory molecules were markedly increased in the CyA injection group, and DA-1229 treatment demonstrated dramatic improvement (Figure 4). However, there were no significant changes in GLP-1 receptor expression among groups.

### 3.5. Effects of DA1229 on Renal Structural Change

Figure 5 shows representative renal pathology and immunochemical staining for PAS, F4/80, Masson’s trichrome, and α-SMA in each group of experimental animals. Consistent with the results of proteinuria, proinflammatory, and profibrotic molecule expression analyses, PAS and Masson’s trichrome staining revealed higher extensive tubulointerstitial damage, including interstitial fibrosis and tubular atrophy, in the CyA group (Figure 5A,C). The expression of F4/80 as a macrophage infiltration marker was increased in the CyA group and decreased in animals treated with DA-1229 (Figure 5B). The CyA group exhibited enhanced fibrosis as demonstrated by increased α-SMA accumulation in the tubulointerstitium (Figure 5D). DA-1229 treatment after CyA injection decreased extensive interstitial fibrosis, tubular atrophy, macrophage infiltration, and α-SMA accumulation in the tubulointerstitium. However, DA-1229 treatment did not result in significant renal structural changes and macrophage infiltration in the vehicle injection group. Quantitative analysis of F4/80 staining and tubulointerstitial fibrosis index showed similar results (Figure 6). These results indicate that DA-1229 treatment significantly ameliorates functional and morphological abnormalities induced by CyA injection.

## 4. Discussion

In this study, we demonstrated that DA-1229 results in significant reductions in urinary albumin excretion and macrophage infiltration, and improves renal structural changes such as tubulointerstitial fibrosis and tubular atrophy. We also observed that DA-1229 markedly suppressed TGF-β1 and type I collagen production in the kidney that was induced by cyclosporine. The mechanism of CyA-induced chronic nephrotoxicity is multifactorial. CyA induces renal hypoxia and increases RAS activity, oxidative stress, TGF-β1 production, and renal structural changes such as tubulointerstitial fibrosis [15,19,20]. In the present study, we confirmed the presence of CyA-induced renal injury by detecting significant increases in serum creatinine, renal oxidative stress, inflammatory markers, profibrotic markers, and renal histopathological changes. DA-1229 treatment significantly ameliorated these pathologic changes induced by CyA.

Previously, we observed that DPP-4 expression was highly increased in proximal tubular cells (PTCs) and that DPP-4 activity was markedly upregulated in PTCs by TGF-β1 [10]. Since the major target site of injury in chronic CyA nephrotoxicity is the tubulointerstitium, we investigated the effects of the DPP-4 inhibitor in this model. Although we did not observe increased DPP-4 activity in plasma and kidney tissue in the CyA group, DA-1229 treatment significantly suppressed DPP-4 activity in plasma and kidney tissue in both vehicle and CyA injection groups. These results suggest that DA-1229 was successfully delivered to the kidney and that it systematically controlled DPP-4 activity. This study differs from previous work because we directly measured DPP-4 activity in the kidney, rather than simply measuring DPPIV expression. In the present study, we used an animal model of progressive renal fibrosis without hyperglycemia to elucidate the effects of DPP-4 inhibition independent of GLP-1. As expected, we did not find any significant differences in plasma levels of GLP-1, even in the group that received the DPP-4 inhibitor treatment. Furthermore, there were no changes in the expression of the GLP-1 receptor among groups. This is the first study to demonstrate the beneficial renoprotective effect of the DPP-4 inhibitor independent of the GLP-1 mechanism in cyclosporine nephrotoxicity.

It is of interest that DA-1229 showed renoprotective effects, despite the CyA group did not show increased DPP-4 activity. The reason for the lack of increase in DPP-4 activity in the CyA group is unclear, but we can speculate several possibilities. First, activation of RAS induced by low salt and activation of TGFβ1 induced by CyA may increase DPP-4 activity, but tubulointerstitial injury and associated cell loss in proximal tubule cells (PTCs) is the prominent feature of CyA-induced nephropathy. We observed an excessive loss of cellularity in fibrotic areas in the CyA group. Considering that most of the DPP-4 activity is from proximal tubule cells, a considerable loss of PTCs may mask the increase in DPP-4 activity in the CyA group. Second, DPP- 4 may not be involved in the inhibition of cyclosporin A induced renal damage by DA-1229. There are several reports showing that DPP-4 inhibitors can prevent cardiac and renal injury independent of DPP-4 inhibition. Linagliptin attenuates cardiac dysfunction after myocardial infarction in DPP-4 deficient rats [21]. Teneligliptin also showed anti-oxidant effects in the kidney and aorta in DPP-4 deficient rats [22]. Furthermore, linagliptin showed preventive effects in CKD progression in rats with 5/6 nephrectomy. DPP-4 activity in 5/6 nephrectomized kidney did not show increased levels, but the DPP-4 inhibitor showed renoprotective effects. In addition, linagliptin also showed renoprotective effects in 5/6 nephrectomy in DPP-4 deficient rats [23]. Collectively, these results suggest that DPP-4 inhibitors may provide organ protective effects independent of DPP-4 activity.

We observed that the increased urinary excretion of 8-isoprostane in the CyA group, which reflected increased oxidative stress in the kidney, was attenuated by treatment with DA-1229. We also found significant upregulation of inflammatory molecules, including MCP-1 and TLR4, and associated increased macrophage infiltration in the CyA group, and these inflammatory changes were attenuated by DA-1229 treatment. This result is in line with those of previous reports indicating that TLR4 is activated in CyA nephrotoxicity and that inhibition of TLR4 activity prevents renal fibrosis [24]. Additionally, the activation of TLR4 has been reported to induce oxidative stress and endothelial dysfunction [25]. A previous study reported that CyA induced an increase of reactive oxygen species (ROS) production and activated nicotinamide adenine dinucleotide phosphate (NADPH) oxidase (NOX) [26] and that DPP-4 inhibition decreases oxidative stress in various organs [27,28,29]. Taken together, these results suggest that DPP-4 inhibition decreases oxidative stress and inflammation in the kidney, leading to improved renal function.

We also detected significant increases in urinary excretion of protein and albumin in the CyA group, suggesting glomerular and tubular injury, and found that DA-1229 treatment markedly attenuated proteinuria and albuminuria. These results are consistent with those of previous preclinical and clinical reports that DPP-4 inhibitors reduced albuminuria in diabetic and non-diabetic kidney injury [2,10,30].

In terms of renal fibrosis, previous studies suggested that TGF-β1 stimulates the development of CyA-induced tubulointerstitial fibrosis through the accumulation of extracellular matrix (ECM) protein and inhibition of ECM degradation [14,31]. In addition, TGF-β1 also induces oxidative stress via activation of NADPH oxidase [32]. In this study, we observed that DA-1229 treatment significantly suppressed the expressions of TGF-β1, connective tissue growth factor (CTGF), and type I collagen gene expression. Furthermore, we also observed that DA-1229 treatment decreased extensive interstitial fibrosis and tubular atrophy. These results agree with those of our previous report showing that the DPP-4 inhibitor protects against renal interstitial fibrosis in a mouse model of ureteral obstruction [10]. In conclusion, the results of our study suggest that the DPP-4 inhibitor exhibits renoprotective properties in the treatment of CyA-induced nephrotoxicity. Inhibiting DPP-4 may offer new treatment options to protect patients from CyA-induced nephrotoxicity beyond the simple improvement of glycemic control.

## Figures and Tables

**Figure 1 life-11-00251-f001:**
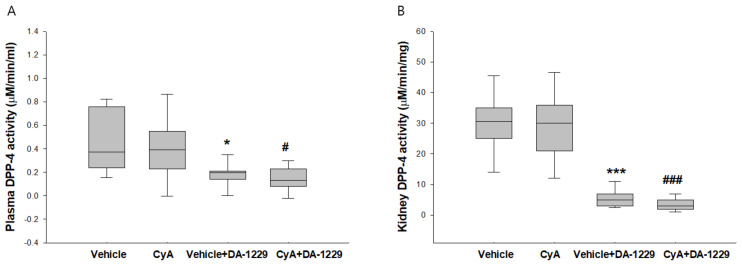
Effects of DA-1229 on DPPIV activity in plasma and kidney tissue at four weeks. (**A**) DPP-4 activity in the plasma. (**B**) DPP-4 activity in the kidney. Data are expressed as median (range). * *p* < 0.05; *** *p* < 0.001 vs. vehicle; ^#^
*p* < 0.05, ^###^
*p* < 0.001 vs. CyA.

**Figure 2 life-11-00251-f002:**
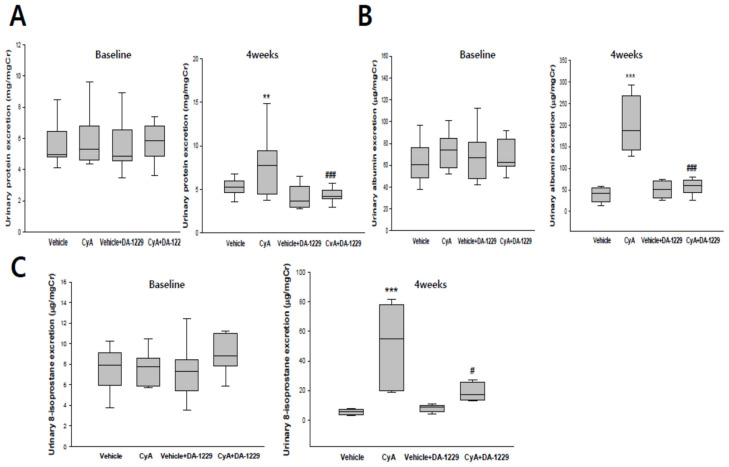
Effects of DA-1229 on urinary excretion of protein, albumin, and 8-isoprostane in experimental animals. (**A**) Twenty-four-hour urinary excretion levels of protein. (**B**) Twenty-four-hour urinary excretion levels of albumin. (**C**) Twenty-four-hour urinary levels of 8-isoprostane. Urinary protein, albumin, and 8-isoprostane levels were corrected using urine creatinine levels. Data are expressed as median (range). ** *p* < 0.01, *** *p* < 0.001 vs. CyA group at baseline period; ^#^
*p* < 0.05, ^###^
*p* < 0.001 vs. the CyA group at 4 weeks.

**Figure 3 life-11-00251-f003:**
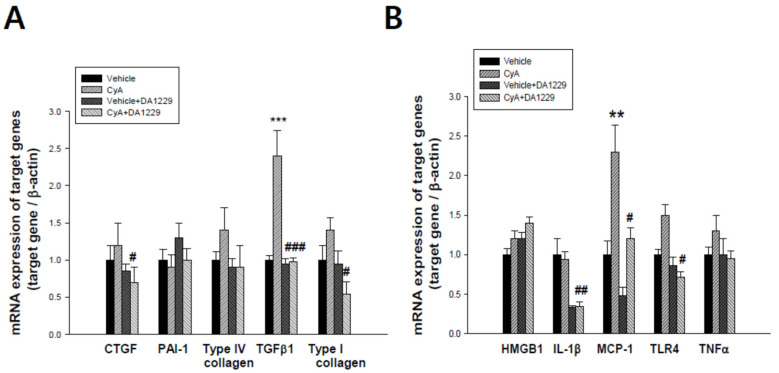
Effects of DA-1229 on inflammatory and profibrotic gene expression at four weeks in experimental animals. (**A**) Effects of DA-1229 on the expression of profibrotic genes. (**B**) Effect of DA-1229 on the expression of inflammatory genes. Data are expressed as mean ± SEM. ** *p* < 0.01, *** *p* < 0.001 vs. vehicle; ^#^
*p* < 0.05, ^##^
*p* < 0.01, ^###^
*p* < 0.001 vs. CyA. CTGF, connective tissue growth factor; PAI-1,plasminogen activator inhibitor-1; TGF-β1, transforming growth factor -β1; HMGB1, high-mobility group box-1; IL-1β, interleukin-1β; MCP-1, monocyte chemoattractant protein-1; TLR4, toll-like receptor 4; TNFα, tumor necrosis factor α.

**Figure 4 life-11-00251-f004:**
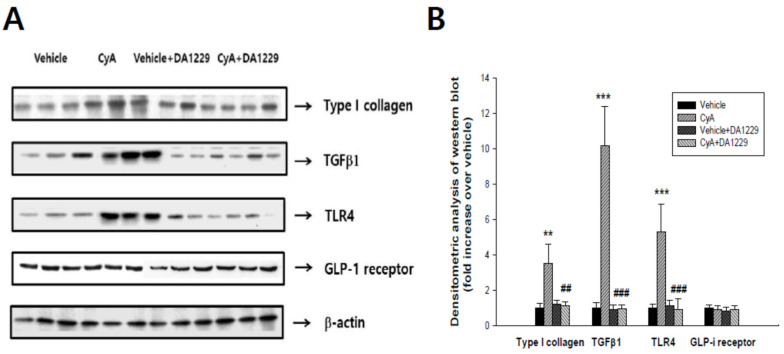
Effects of DA-1229 on the expressions of profibrotic, proinflammatory molecules, and GLP-1 receptor in renal cortical tissues at four weeks in experimental animals. (**A**) Representative western blots for Type I collagen, TGF-β1, TLR4, and GLP-1 receptor in the kidney at 4 weeks. (**B**) Densitometric analysis of western blot results. Data are expressed as mean ± SEM. ** *p* <0.01, *** *p* < 0.001 vs. vehicle; ^##^
*p* < 0.01, ^###^
*p* < 0.001 vs. CyA.

**Figure 5 life-11-00251-f005:**
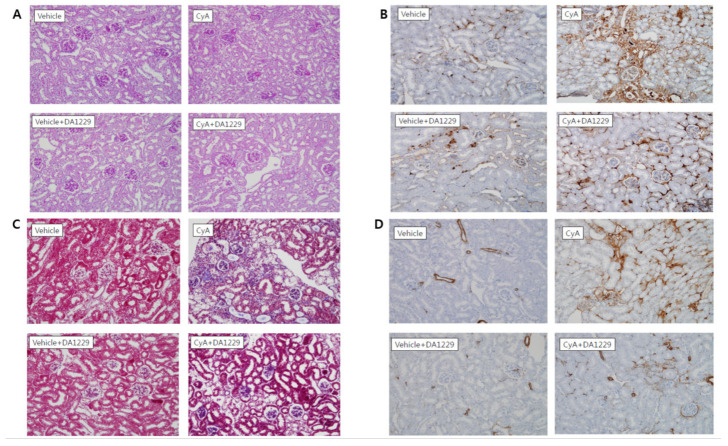
Representative renal pathology and immunohistochemistry in experimental animals at 4 weeks. (**A**) PAS stain. (**B**) F4/80 stain. (**C**) Masson’s trichrome stain. (**D**) α-smooth muscle actin (α-SMA) stain. Original magnification × 400.

**Figure 6 life-11-00251-f006:**
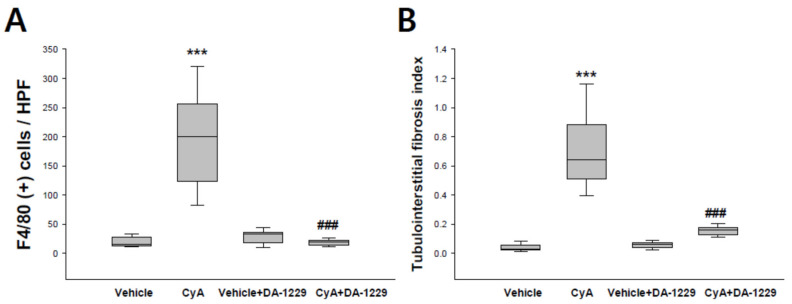
Quantitative analysis with immunohistochemical staining. (**A**) Effects of DA1229 on macrophage infiltration. (**B**) Effects of DA-1229 on tubulointerstitial fibrosis. Data are expressed as median (range). *** *p* < 0.001 vs. vehicle; ^###^
*p* < 0.001 vs. CyA.

**Table 1 life-11-00251-t001:** Physical and biochemical parameters of experimental animals.

Parameters	Week	*Vehicle*	*CyA*	*Vehicle + DA1229*	*CyA + DA1229*
Number	4	10	10	10	10
Body weight (g)	0	33 (32–35)	33 (31–34)	33 (32–35)	33 (32–38)
	4	37.5 (33–39)	36 (33–38)	38 (37–43)	38.5 (35–44)
Food intake (g/day)	0	5.5 (5.1–6)	5.0 (4.7–5.3)	4.4 (4.2–5.1)	5.1 (4.6–5.6)
	4	4.6 (4.1–5.7)	3.0 (2.3–3.2)	5.2 (4.4–5.9)	4.7 (4.1–5.6)
Water intake (g/day)	0	5.4 (5.1–5.7)	4.5 (4.4–5.1)	4.0 (3.2–4.2)	5.8 (4.8–7.1)
	4	5.2 (3.1–6.5)	5.3 (2.5–7.7)	6.9 (5.5–8.1)	8.5 (6.6–11.1)
Urine volume (mL/day)	0	3.2 (2.6–4.1)	3.3 (2.3–4.7)	4.1 (2.9–4.7)	3.2 (2.3–4.1)
	4	3.1 (2.6–4.1)	3.1 (1.9–3.6)	3.1 (1.9–5.1)	3.5 (2.3–4.2)
Serum Cr (μmol/L)	4	46 (35–53)	76 (57–82) ***	48 (41–59)	47 (39–52) ^###^
Active GLP-1 (pM)	4	2.25 (1.56–3.14)	2.12 (0.98–4.21)	2.02 (1.11–4.12)	2.22 (0.76–5.12)
SBP (mmHg)	4	95 (79–111)	98 (89–116)	92 (81–110)	102 (98–110)

Cr, creatinine; GLP-1, glucagon like peptide-1; SBP, systolic blood pressure. Values are expressed as median (range). *** *p* < 0.001 vs. vehicle group; ^###^
*p* < 0.001 vs. CyA group.

## Data Availability

Not applicable.

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
