# Peer review of "A Novel Dipeptidyl Peptidase-4 Inhibitor DA-1229 Ameliorates Tubulointerstitial Fibrosis in Cyclosporine Nephrotoxicity in Mice"

_life, 2021, doi:10.3390/life11030251_

Round 1

Reviewer 1 Report

This is an observational paper describing work to investigate the effects of administrating DA-1229, a DPP-4 inhibitor, on cyclosporin A induced chronic nephrotoxicity.

The work shows that DPP-4 inhibits cyclosporin A induced renal damage but does not describe the biological pathways involved in this inhibition. The authors point out that cyclosporin A induced renal damage is multifactorial and so defining the pathways driving the renal damage and inhibition by DDP4 inhibitor would require substantially more work than is described here.

A confusing result is the lack of increase in DPP4 in plasma and kidney with cyclosporin A administration with the concurrent increase in markers of renal damage (urinary protein, albumin and 8-isoprostane), fibrosis (CTGF, TGF-b and type 1 collagen) and target genes (IL-1b, MCP-1 and TLR4) and the subsequent decrease in DPP-4 with administration of DA-1229 and the lack of change in the markers compared to the vehicle control. Together these observations suggest either; DPP4 is not involved in inhibition of cyclosporin A induced renal damage by DA-1229 or that the DPP4 is inhibited in a location other than the blood or kidney and that this inhibition has an effect in the kidney environment. It would have been good to know the authors thoughts about how the administration of DA-1299 had an effect on the markers of fibrosis and target genes without changing the levels of DPP4 in the blood or kidney.

That said the work is of good quality and well described except for the method measuring DPP4 activity. I cannot find this method in the text, this is an important omission and should be rectified, since the authors point out that this work differs from their previous work because they ‘directly measured DPP-4 activity in the kidney, rather than simply measuring DPPIV expression.’  

Author Response

This is an observational paper describing work to investigate the effects of administrating DA-1229, a DPP-4 inhibitor, on cyclosporin A induced chronic nephrotoxicity. The work shows that DPP-4 inhibits cyclosporin A induced renal damage but does not describe the biological pathways involved in this inhibition. The authors point out that cyclosporin A induced renal damage is multifactorial and so defining the pathways driving the renal damage and inhibition by DDP4 inhibitor would require substantially more work than is described here.

à Thank you for your comments. In this manuscript, we have shown that CyA administration induces renal injury via increase in inflammation, fibrosis and oxidative stress. These results are in agree with many previous reports in CyA-induced nephropathy. However, interesting finding is that DPP-4 inhibitor showed the renoprotective effects by suppressing these molecular changes induced by CyA. DPP-4 inhibitor showed renoprotective effects, despite CyA group did not show increased DPP-4 activity in this paper. The reason for lack of increase in DPP-4 activity in CyA group is unclear, but we can speculate several possibilities. First, activation of RAS induced by low salt and activation of TGFb1 induced by CyA may increase DPP-4 activity, but tubulointerstitial injury and associated cell loss in proximal tubule cells (PTCs) is the prominent feature of CyA-induced nephropathy. We observed that excessive loss of cellularity in fibrotic areas in CyA group. Considering that most of DPP-4 activity is from proximal tubule cells, considerable loss of PTCs may mask the increase in DPP-4 activity in CyA group. Second, DPP- 4 may not be involved in inhibition of cyclosporin A induced renal damage by DA-1229. There are several reports showing that DPP-4 inhibitors can prevent cardiac and renal injury independent of DPP-4 inhibition. Linagliptin attenuates cardiac dysfunction after myocardial infarction in DPP-4 deficient rats (Yamaguchi T et al, J Pharmacol Sci. 139(2):112-119, 2019). Teneligliptin also showed anti-oxidant effects in the kidney and aorta in DPP-4 deficient rats (Kimura S et al, Metabolism. 65(3):138-145, 2016). Furthermore, linagliptin showed preventive effects in CKD progression in rats with 5/6 nephrectomy. DPP-4 acivity in 5/6 nephrectomized kidney did not show increased levels, but DPP-4 inhibitor showed renoprotective effects. In addition, linagliptin also showed renoprotective effects in 5/6 nephrectomy in DPP-4 deficient rats (Tsuprykov O et al, Kidney Int. 89(5):1049-1061, 2016). Collectively, these results suggest that DPP-4 inhibitor may provide organ protective effects independent of DPP-4 acitivity. We added these important points in Discussion section in revised manuscript.

A confusing result is the lack of increase in DPP4 in plasma and kidney with cyclosporin A administration with the concurrent increase in markers of renal damage (urinary protein, albumin and 8-isoprostane), fibrosis (CTGF, TGF-b and type 1 collagen) and target genes (IL-1b, MCP-1 and TLR4) and the subsequent decrease in DPP-4 with administration of DA-1229 and the lack of change in the markers compared to the vehicle control. Together these observations suggest either; DPP4 is not involved in inhibition of cyclosporin A induced renal damage by DA-1229 or that the DPP4 is inhibited in a location other than the blood or kidney and that this inhibition has an effect in the kidney environment. It would have been good to know the authors thoughts about how the administration of DA-1299 had an effect on the markers of fibrosis and target genes without changing the levels of DPP4 in the blood or kidney.

à We agree with this important point. In terms of lack of increase in DPP-4 activity with increased markers of renal damage in CyA group, we have suggested the possible explanation for lack of increase in DPP-4 activity in CyA group in previous response part. CyA administration induces severe renal injury in our experiment similar with many previous reports. In this experiment, DPP-4 activity in plasma and kidney was markedly decreased after administration with DA-1229, and these results suggest that DA-1229 was successfully delivered to the kidney and that it systematically controlled DPP-4 activity. Since CyA administration induces severe renal injury, the beneficial effects of DA-1229 in this experiment is from direct effects on kidney rather than systemic effects. Reviewer 1 pointed out important points that DPP- 4 may not be involved in inhibition of cyclosporin A induced renal damage by DA-1229. We agree with these points and we already described in previous response part in this important issue, and we added these important points in Discussion section in revised manuscript.

That said the work is of good quality and well described except for the method measuring DPP4 activity. I cannot find this method in the text, this is an important omission and should be rectified, since the authors point out that this work differs from their previous work because they ‘directly measured DPP-4 activity in the kidney, rather than simply measuring DPPIV expression.’  

à As reviewer 1 suggested, we added clearly methods for measurement of DPP-4 activity in the Method section in revised manuscript.

The corrections which we made were shown in underlined fonts and highlighted with yellow color in revised manuscript. Again thank you very much for your excellent reviewing of this manuscript and I look forward to hearing your decision soon.

Reviewer 2 Report

The study and its findings are interesting. The authors showed that a DDP-4 inhibitor, a compound that belongs to an already used and safe medication class, protects the mouse kidney from CsA-induced nephrotoxicity. The study is well-conducted, and the manuscript well-written.

I have the following comments:

  1. In the introduction. Although CNI-toxicity was thought to play a major role in chronic kidney allograft failure, nowadays, it is accepted that chronic antibody-mediated rejection is responsible for most cases. A small paragraph about it would be well-come since it would place the significance of the evaluated problem in its real dimensions.
  2. How the doses of the DDP-4 inhibitor and CsA were selected?
  3. The statistical analysis section needs to become more accurate. The authors wrote, “Multiple comparisons were carried out using Wilcoxon’s rank-sum tests and Bonferroni correction. The Kruskal–Wallis test was used to compare more than two groups, followed by the Mann–Whitney U-test.” Which of the two?
  4. When non-parametric tests are used, median (IQR) is more proper than mean (SEM) to express and depict the results. Please revise accordingly.
  5. In the discussion, it is written that TGF-β1 increases DDP-4 expression. In the study, CsA increases TGF-β1. However, DDP-4 activity remained stable. A comment would be well-come.

Author Response

The study and its findings are interesting. The authors showed that a DDP-4 inhibitor, a compound that belongs to an already used and safe medication class, protects the mouse kidney from CsA-induced nephrotoxicity. The study is well-conducted, and the manuscript well-written. I have the following comments:

  1. In the introduction. Although CNI-toxicity was thought to play a major role in chronic kidney allograft failure, nowadays, it is accepted that chronic antibody-mediated rejection is responsible for most cases. A small paragraph about it would be well-come since it would place the significance of the evaluated problem in its real dimensions.

à Thank you for your comments. We absolutely agree with this point, and we added these clinically important points in the Introduction section in revised manuscript.  

  1. How the doses of the DDP-4 inhibitor and CsA were selected?

à We selected the dose and administration route of DA-1229 and CyA based on a previous study in mice. We added this points in the Material and Method section in revised manuscript.

  1. The statistical analysis section needs to become more accurate. The authors wrote, “Multiple comparisons were carried out using Wilcoxon’s rank-sum tests and Bonferroni correction. The Kruskal–Wallis test was used to compare more than two groups, followed by the Mann–Whitney U-test.” Which of the two?

à First of all, we would like to apologize for this mistake. We corrected this part in statistical analysis section in revised manuscript.

  1. When non-parametric tests are used, median (IQR) is more proper than mean (SEM) to express and depict the results. Please revise accordingly.

à As reviewer 2 pointed out, we changed the results in Table1, Figure 1, 2, and 6 as median (range) in revised manuscript.

  1. In the discussion, it is written that TGF-β1 increases DDP-4 expression. In the study, CsA increases TGF-β1. However, DDP-4 activity remained stable. A comment would be well-come.

à We agree with this important point, we added possible explanation in the Discussion section in revised manuscript.

We highlighted the changes with yellow color in underlined fonts. We would like to thank the reviewer for excellent reviewing of this manuscript and I look forward to hearing your decision soon.

Round 2

Reviewer 2 Report

The authors covered the raised issues adequately.